# Silicon Reduce Structural Carbon Components and Its Potential to Regulate the Physiological Traits of Plants

**DOI:** 10.3390/plants14121779

**Published:** 2025-06-11

**Authors:** Baiying Huang, Danghui Xu, Wenhong Zhou, Yuqi Wu, Wei Mou

**Affiliations:** College of Ecology, Lanzhou University, No. 222, South Tianshui Road, Lanzhou 730000, China; hby1230214@163.com (B.H.); zhouwh2024@lzu.edu.cn (W.Z.); wuyq2024@lzu.edu.cn (Y.W.); mmmuwei@163.com (W.M.)

**Keywords:** Tibetan Plateau, alpine grassland, nutrient absorption, structural carbon components, photosynthesis, stoichiometry

## Abstract

Phosphorus (P) and silicon (Si) could profoundly affect the net primary productivity (ANPP) of grassland ecosystems. However, how ecosystem biomass will respond to different Si addition, especially under a concurrent increase in P fertilization, remains limited. With persistent demand for grassland utilization, there is a need to enhance and sustain the productivity of grasslands on the Qinghai–Tibet Plateau. Three P addition rates (0, 400, 800, and 1200 kg Ca(H_2_PO_4_)_2_ ha^−1^ yr^−1^) without Si and with Si (14.36 kg H_4_SiO_4_ ha^−1^ yr^−1^) were applied to alpine grassland on the Qinghai–Tibet Plateau to evaluate the responses of aboveground biomass and the underlying mechanisms linking to structural carbon composition and physiological traits of grasses and forbs. Our results show that the application of Si significantly reduced the lignin, cellulose, hemicellulose, and total phenol contents of both grasses and forbs. Additionally, the addition of P, Si, and phosphorus and silicon (PSi) co-application significantly increased the net photosynthetic rate (Pn) and light use efficiency (LUE) of grasses and forbs. Moreover, Si promoted the absorption of N and P by plants, resulting in significant changes in the Si:C, Si:P, and Si:N ratios and increasing the aboveground biomass. Our findings suggest that Si can replace structural carbohydrates and regulate the absorption and utilization of N and P to optimize the photosynthetic process of leaves, thereby achieving greater biomass. In summary, Si supplementation improves ecosystem stability in alpine meadows by optimizing plant functions and increasing biomass accumulation.

## 1. Introduction

In recent years, atmospheric nitrogen (N) deposition caused by human activities has significantly increased N inputs in ecosystems, and N enrichment has changed the nutrient restriction mode of ecosystems from N limitation to phosphorus (P) limitation [1,2]. Phosphorus limitation, thus threatening the growth and survival of plants. P, a macrotrophic element of plants and a structural component of nucleic acids and membrane lipids, plays a crucial role in primary and secondary metabolism [3]. P deficiency severely limits the production of plant biomass, changes metabolic pathways, and reduces chlorophyll levels and photosynthetic efficiency [4,5].

Si is the second most abundant element in Earth’s crust and soil, and Si is also considered a beneficial element for regulating plant growth [6]. Most plants can actively or passively absorb soluble silicic acid from soil solution and accumulate it in tissue [6]. Furthermore, the absorption of Si by plants can be divided into three major categories: active, passive, and repulsive, which are, respectively, closely related to plants with high, medium, and low Si accumulation [7]. Both the Poaceae and Cyperaceae families are high Si accumulation plants, containing much higher Si concentrations than other families [8,9]. In plants, Si is associated with many positive physiological responses and has been shown to increase yield, quality, and stress resistance. However, only plants with efficient Si transportation systems (such as Poaceae and Cyperaceae) can benefit more from Si.

At present, the use of Si in alleviating P limitations has received much attention. P restriction exists in many soils because a large part of soil P is stored in forms that plants cannot obtain, such as combining with iron minerals or stable organic matter [10]. However, the laboratory experiments of Jorg Schaller confirmed that Si addition could significantly increases P mobilization, by mobilizing Fe(II)-P phases from mineral surfaces, which increased the availability of P in the soil [11]. Furthermore, studies have shown that Si can optimize the absorption, assimilation, and distribution of N and P in plants [12] and change the C:N:P stoichiometric ratio [13]. Meanwhile, the availability of Si can enhance the active uptake of P by roots through upregulating the expression of P transporter genes, thereby increasing P concentration in plants [14]. For example, Silke Neu et al. found that with the increase in Si supply, the P concentration in winter wheat (*Triticum aestivum* L.) plants increased from slight restriction to optimal nutrition [13]. Moreover, studies have shown that the concentrations of Si and P in plants (such as reeds and wheat) are positively correlated [12,15].

Si plays a significant role in the nutrient cycle. In addition to improving the nutrient stoichiometric coefficients in leaves, Si also affects the concentration of carbon (C)-based defense compounds in plants, such as cellulose, phenols, and lignin [16]. Negative relationships between leaf Si and concentrations of C-based compounds have also been reported (e.g., lignin, cellulose), which has reinforced the contention of a mechanical role of silicification and has led to suggesting ‘trade-offs’ between Si and C components in leaves [17,18]. In a previous single-species experiment with *Phragmites australis* [19], it was confirmed for the first time that Si can replace the corresponding proportion of plant C. Moreover, Si also interacts with structural carbon in plants [20]. It has been suggested that Si may be an energy-cheap alternative to energy-expensive structural compounds [21]. The synthesis of structural carbon compounds (e.g., lignin and cellulose) requires 10 to 20 times more energy than the incorporation of structural SiO_2_ (by material weight) [22]. However, de Tombeur et al. [23] believe that the cost of silicification should be higher than currently recognized. This is due to the existence of overlooked direct costs related to silicon accumulation in leaves (such as the cost related to the movement of soil Si before absorption by plants), which should also be estimated in future studies. Many studies have shown that whether Si increases or decreases, plant cellulose content and lignin content varies according to species and plant tissue function [24]. In addition, the changes in the structural carbon composition (lignin, cellulose) caused by Si will improve the upright character of leaves and change the canopy structure, thereby improving photosynthesis and promoting dry matter accumulation [25]. Moreover, possibly due to the similar functions of phenols and Si on pathogens, Si content appears to be negatively correlated with plant phenol content [20].

Plant productivity in alpine grasslands is declining in response to global climate change and anthropogenic disturbances [26,27]. Therefore, it is necessary to take scientific management measures to improve and maintain grassland productivity, which is critical for protecting animal husbandry and ecological security on the Qinghai–Tibet Plateau. Given its positive role in alleviating biotic and abiotic stresses in crops, Si can be used as a beneficial element in conjunction with phosphate fertilizer. However, most of the previous research studies focused on crop ecosystems or Si-added field manipulative experiments, while less considered the grassland ecosystems [28,29]. Therefore, the aim of this study was to elucidate the responses of adding P and Si to the structural carbon composition and photosynthetic performance of grass and forb in alpine grasslands. Our core hypotheses are as follows: (1) Si can replace plant structural carbon components, reduce the synthesis of structural carbon components. (2) The addition of P and Si can improve plant photosynthesis and promote plant growth. (3) The addition of P and Si can promote the absorption of N and P and the accumulation of C in plants, thus increasing aboveground biomass.

## 2. Results

### 2.1. Structural Carbon Components and Phenol Content

Compared with the control, the addition of P alone did not alter the contents of cellulose, lignin, hemicellulose, and total phenol in the grasses. In contrast with increasing P application rates, the hemicellulose content in forbs progressively increased (up to 4%), while total phenol content showed a concomitant decrease (down to 1%). In addition, no significant interactions between P and Si were observed on lignin, cellulose, hemicellulose, and total phenol content in grasses (Figure 1A–D).

With the addition of Si alone, the contents of lignin, cellulose, hemicellulose, and total phenol in grasses decreased by 28%, 30%, 23%, and 25%, respectively. In addition, significant interactions between P and Si were observed only for the lignin (*p* = 0.011), hemicellulose (*p* = 0.015), and total phenol contents (*p* = 0.009) of forbs (Figure 1E–H). Similar to grass, with the addition of Si alone, the contents of lignin, cellulose, hemicellulose, and total phenol in forbs decreased by 29%, 15%, 18%, and 38%, respectively. However, compared with Si alone, the addition of P significantly increased the lignin content and decreased the hemicellulose and total phenol contents. In general, the contents of lignin, hemicellulose, and total phenol in grasses were greater than those in forbs, which was not related to the supply of P and Si.

### 2.2. Photosynthetic Performance

With increasing P application rates, the Pn, Gs, Tr, and LUE of grasses and forbs progressively increased. The Pn, Tr, WUE, and LUE of grasses were higher than those of forbs. Compared with the control, the addition of Si alone increased the Pn, Gs, Ci, WUE, and LUE of grass but significantly decreased the Pn, Gs, Ci, WUE, and LUE of forbs. The Pn (ns; *p* < 0.001), Gs (*p* = 0.01; *p* < 0.001), Tr (*p* < 0.001; ns), Ci (*p* = 0.006; ns), WUE (*p* < 0.001; *p* < 0.001), LUE (*p* = 0.031; P < 0.001) of grasses and forbs was significantly correlated with the interaction of P × Si supplementation (Figure 2). At all P levels, the Pn and LUE of grasses with added Si significantly increased compared with those without added Si, where the highest values of these parameters were reached under the P3 treatment. Under the P2 treatment, the addition of Si significantly increased the Pn, Gs, Tr, and LUE of forbs, and the photosynthetic rate reached a maximum of 18.52 μmol·m^−2^·s^−1^. However, under the CK and SiP3 treatments, the incorporation of Si reduced the Pn of forbs by 15% and 6%, respectively (Figure 2).

### 2.3. Stoichiometric Ratios and Biomass

Compared with the control, the addition of P alone significantly increased the Si:C (up to 7%) and Si:N (up to 7%) ratios of grasses but significantly decreased the Si:P (down to 50%) ratio of grasses and the Si:C (down to 41%), Si:N (down to 37%), and Si:P (up to 74%) ratios of forbs. Compared with those of the control group, the Si:C ratios of grass and forb leaves increased by 37% and 40%, the Si:N ratios of grass and forb leaves increased by 35% and 44%, and the Si:P ratio of forb leaves decreased by 10%. Two-factor ANOVA revealed a very significant interaction effect between P and Si on the Si:C (ns; *p* < 0.001), Si:N (*p* < 0.001; *p* < 0.001) and Si:P (ns; *p* < 0.001) ratios of grasses and forbs (Figure 3). At all P levels, the Si:C and Si:N ratios of grasses were significantly increased by the addition of Si, but the Si:N ratio was significantly lower than that resulting from the addition of Si alone by approximately 7%, and the effect on the Si:P ratio was not significant. Similarly, with increasing P addition amount, the incorporation of Si significantly increased the Si:C and Si:N ratios and decreased the Si:P ratio of forbs. Moreover, under the P1 treatment, the incorporation of Si reduced the forb Si:P ratio by up to 50%.

The biomass of grasses and forbs increased significantly with increasing P supplementation. Similarly, compared with the control, Si application alone increased the biomass of grasses and forbs by 8% and 14%, respectively. There was a significant interaction effect between P and Si levels on grass and forb biomass (*p* < 0.001; *p* < 0.001) (Figure 4). The biomass under the P × Si treatment was significantly higher than that under Si or P addition alone. Under the P3Si treatment, the biomass of grasses and forbs reached maximum values of 819 and 967 g.cm^−2^, respectively.

### 2.4. Correlation and Principal Component Analysis

The correlation index results for grasses revealed that the aboveground biomass were positively related with N, P, and Si contents; negatively correlated with lignin, cellulose, hemicellulose, and total phenol contents; and positively correlated with Pn, Gs, Tr, Ci, and LUE (*p* < 0.05) (Figure 5A). The correlation of forb indexes showed that aboveground biomass was positively correlated with C, N, and P contents (*p* < 0.05); negatively correlated with lignin, cellulose, hemicellulose, and total phenol contents; and positively correlated with Pn, Gs, Tr, and LUE (*p* < 0.05) (Figure 5B).

PCA was used to analyze the plant nutrients (C, N, P, and Si), aboveground biomass, photosynthetic parameters, and structural carbon components of the two functional groups under the addition of P and Si (Figure 6). In the PCA of grasses and forbs, the first two components explained 79.7% and 77.7% of the total variance, respectively (Figure 6A,B). For the two plant functional groups, the separation of the Si-added and non-Si-added treatments was obvious. Si, LUE, and Gs were the main explanatory variables of PC1, while C, WUE, Tr, Pn, and aboveground biomass contributed the most to PC2, C, N, P, and aboveground biomass in forbs contributed the most to the variation in the PC1 explanatory effect, while Si, Ci, and Pn were the main explanatory variables in PC2.

## 3. Discussion

### 3.1. Silicon Decreased Plant Structural Carbon Components and Phenols

To determine the effect of Si amendment on the carbon composition of plant leaf structures under different P levels, we determined the contents of cellulose, lignin, and hemicellulose in the leaves. We observed that the addition of Si significantly reduced the lignin, cellulose, and hemicellulose contents in grass and forb leaves (Figure 1), and these results strongly support the idea that Si can replace structural carbon compounds. In addition, the availability of Si reduced the phenol content in grass and forb leaves (Figure 1D,H), possibly due to the similar functions of phenols and Si on pathogenic bacteria [30]. Structural carbon composition has been reported to vary widely by genus and plant species [31,32]. The results showed that the contents of lignin, hemicellulose, and total phenols in grasses were greater than those in forbs. Plants grown without Si have droopy leaves and smooth surfaces, while plants grown with Si have upright leaves and rough surfaces. Tamai and Ma [33] reported that Si plays a beneficial role in improving plant bedding resistance and increasing leaf upright orientation, which leads to better light transmission through the plant canopy, thereby indirectly improving the photosynthesis of the whole plant, especially for shorter plant species [34].

### 3.2. Silicon Increased Plant Photosynthesis

Photosynthesis plays a crucial role in the growth and development of plants. This study found that in all treatments, the Pn, Tr, WUE, and LUE of grasses were higher than those of forbs. This might be due to the fact that in alpine ecosystems, grasses are taller than nongrasses [35,36], which favors grasses when competing for light. The application of sufficient amounts of P fertilizer can increase the Gs of plants and thus enhance their photosynthetic capacity [37]. In this study, consistent with previous studies, appropriate P addition significantly increased the Gs and Pn values of grasses and forbs (Figure 2A,B,G,H).

Ahmed et al. [38] reported that the application of exogenous Si increased the Si content in leaves (Appendix A), and Si was deposited in the cell wall in the form of amorphous silica (SiO_2_.nH_2_O), forming a double layer of silica-cuticle double layer which has a good osmotic regulation effect. Thus, the photosynthetic capacity of plants was improved [39,40]. Previous studies have shown that the application of an appropriate amount of Si can significantly improve the net photosynthetic rate, transpiration rate, and stomatal conductance of plants [41,42]. Moreover, studies have shown that the application of exogenous Si can enhance stomatal conductance, thus promoting the photosynthesis of crops [43]. For example, Gao et al. [44] found that Si can act as a regulator of stomata in maize leaves and promote photosynthesis. The results of this study are basically consistent with previous research results. In the grass functional group, the Pn, Tr, Ci, and LUE of plants increased under different P application levels (Figure 2). Similar results were observed in forb plants in the P1Si and P2Si treatments.

Interestingly, in the Si and P3Si treatments, the Pn of forb plants was significantly reduced. This might be due to the fact that silicon indirectly reduces the light and nutrient acquisition of forbs by enhancing the competitiveness of species (such as accelerating canopy closure) [45]. Furthermore, studies have shown that Ci is influenced by factors such as light intensity and stomatal conductance [37,46]. Therefore, we speculate that the P3Si treatment reduced leaf Gs and Tr, potentially leading to the accumulation of Ci. The excessive Ci could not be consumed, thus limiting photosynthesis (Figure 2G–J). Second, the addition of Si alone significantly reduced the photosynthesis of forb plants mainly due to stomatal limitation. Stomatal conductance is one of the important limiting factors affecting the net photosynthetic rate. When stomatal conductance decreases, the amount of CO_2_ entering the stomata is reduced, which cannot meet the needs of photosynthesis [47]. This explains the reason for the decrease in the net photosynthetic rate of forbs under the Si and P3Si treatments.

### 3.3. Si Increased Biomass and Changed Plant Stoichiometry

Plants can adopt different strategies to adapt to environmental changes; different functional groups respond differently to nutrient environments, and their nutrient distribution patterns also present different ecological stoichiometric characteristics [48,49]. Grasses absorb more Si than forbs because they have proteins in their root tissue membranes dedicated to active Si absorption and transport [50]. P addition increased above-ground biomass and P concentration, resulting in a decrease in the plant Si:P ratio, which was also reported in previous studies [51,52] (Figure 3C,F, Figure 4, and Appendix A). Furthermore, we observed that P addition alone significantly increased Si content, Si:C, and Si:N ratios in grasses, while markedly decreasing these parameters in forbs (Figure 3 and Appendix A). This phenomenon likely results from the higher affinity of P for adsorption sites on iron/aluminum (Fe/Al) oxide surfaces in acidic soils, which reduces Si desorption [53]. Graminoids possess specialized Si transporters—Lsi1 (low-affinity influx channel) and Lsi2 (active efflux transporter) that enable efficient Si uptake [50]. Thus, despite P-induced decreases in soil Si availability, grasses maintain elevated tissue Si through these dedicated transport systems and root adaptations [54]. In contrast, forbs (non-Si accumulators), which lack such transporters and rely on passive apoplastic diffusion, exhibit Si absorption rates linearly correlated with soil solution Si concentration [50]. Consequently, P application reduces foliar Si in forbs. These divergent responses between Si-accumulating grasses and non-accumulating forbs suggest that PSi interactions may indirectly shape plant community composition.

Principal component analysis (PCA) revealed that increased Si content in the leaves of both forbs and grasses was synchronized with aboveground biomass accumulation (Figure 6). This demonstrates that Si acts as a beneficial element for biomass production, consistent with prior studies [14,15]. Silicon addition significantly elevated both soil total P and available P content (Appendix A). Concurrently, the addition of Si significantly increased the P content and Si:C and Si:N ratios of grasses and forbs, and decreased the Si:P ratio of forbs (Figure 3 and Appendix A). Research by Hu et al. demonstrated that silicates promote soil P release, while Si enhances plant P uptake efficiency, thereby improving P bioavailability [55]. For instance, consistent with the present results, foliar Si fertilizer spraying increased the P content in non-Si-accumulating (*Moringa oleifera* L.) plants [56], resulting in a decrease in Si:P. The increase in P content in grasses and forbs may be through increasing the elongation of stems or the number of branches, increasing the plant height or cover area to obtain more light resources and promote the accumulation of biomass. In addition, our results show that increasing Si supply significantly reduces the structural carbon component (Figure 1), suggesting that there should be a negative correlation between Si and C [57]. However, the correlation between Si and C in grasses and forbs was not significant (Figure 5), suggesting that carbon saved from lignin substitution by Si is reallocated to photosynthesis. This enhances the Pn and subsequently increases photosynthetic assimilate allocation to growth (Figure 2). We found a synergistic relationship between Si and N in grasses and forbs. Carey and Fulweiler [58] also showed that N availability is positively correlated with Si accumulation. The effect of Si on the nutrient ratio of grasses is attributable to the effects of Si on N and P, while in forbs, the effect of Si on N is responsible (Figure 5).

The effects of Si on grasses and forbs are not limited to increasing the plant’s photosynthetic or structural compounds. As noted above, Si also regulates elemental stoichiometric homeostasis, confirming a biological strategy reported by Hao et al. [59] in other forages. Therefore, the knowledge gained from this study is critical for developing optimal nutrition management strategies. Changing the Si:C, Si:N, and Si:P mass ratios and structural carbon compounds of plants may have a chain effect on nutrient levels by affecting pasture quality [60]. In forage grasses, cellulose, hemicellulose, and lignin constitute are the core components that determine the mechanical resistance of plant fibers. Studies have shown that fiber and total phenolic compounds are the primary drivers of palatability, where higher fiber content may increase toughness and resistance to herbivore mastication, consequently reducing the palatability of these plants [61]. However, in this study, silicon supply was found to reduce lignin, cellulose, and hemicellulose content in both grasses and forbs (Figure 1). This suggests that silicon fertilizer application can result in softer forage texture, improved palatability, and enhanced livestock intake and digestibility, thereby increasing forage quality and utilization efficiency [62]. To summarize, Si optimizes plant structural carbon compounds, enhances P and N uptake efficiency, and refines physiological metabolism and ecological strategies. These synergistic effects ultimately increase biomass production in both grasses and forbs while improving forage quality.

## 4. Materials and Methods

### 4.1. Study Location

The research area is located at the Alpine Meadow and Wetland Ecosystem Positioning Research Station of Lanzhou University (Azi Branch Station) in Maqu County, Gannan Tibetan Autonomous Prefecture, Gansu Province, China (33°39′ N, 101°53′ E, 3650 m), which is located on the eastern margin of the Qinghai–Tibet Plateau (Figure 7). The climate is cold and humid, with an average annual temperature of 1.1 °C and an average annual precipitation of 615 mm [63]. The area has an alpine, semihumid, and semiarid climate, with rainfall concentrated in July and August in summer. The annual sunshine duration is approximately 2580 h, and the annual frost period is more than 172 d. The vegetation type is alpine grassland, and the dominant species include *Carex myosuroides* Vill, *Elymus nutans* Griseb, *Agrostis hugoniana* Rendle, *Poa pratensis* L, *Ranunculus tanguticus* (Maxim.) Ovcz, and *Anemone rivularis var. flore-minore* Maxim. The soil type is subalpine meadow soil. The specific average annual precipitation and temperature data at the experimental site from 2012 to 2020 are presented in Appendix A.

### 4.2. Experimental Design

The experimental site is located in a forbidden grazing area and is protected by a fence to prevent cattle and sheep from trampling and feeding. Using a completely random block design, we selected a plot with relatively flat terrain and uniform texture among typical natural grasslands in early May 2012 for the experiment. The experiment was a two-factor experiment involving P addition and Si addition and consisted of eight treatments with four phosphate fertilizer dosages denoted CK, P1, P2, and P3 (0, 49, 98, and 148 kg P ha^−1^ yr^−1^ or 0, 400, 800, and 1200 kg Ca(H_2_PO_4_)_2_ ha^−1^ yr^−1^) and two Si dosages corresponding to a silicon-free (Si-) treatment and silicon addition (Si+, 14.36 kg H_4_SiO_4_ ha^−1^ yr^−1^); the final treatments were denoted CK, P1, P2, P3, Si, P1Si, P2Si, and P3Si. There were 6 replicates per treatment, for a total of 48 quadrats, each of which was 2 m×2 m long, separated by 2 m, and fertilized in May each year. Before P and Si addition, the soil nutrient content is shown in Appendix A.

### 4.3. Determination of Photosynthetic Parameters

In mid-August 2020, in vivo measurements of photosynthetic parameters were performed on intact leaves prior to sample collection. The net photosynthetic rate (Pn; μmol m^−2^ s^−1^), intercellular CO_2_ concentration (A; μmol CO_2_ m^−2^ s^−1^), stomatal conductance (Gs; μmol m^−2^ s^−1^), and transpiration rate (E) were measured. An illuminometer was used to measure the light intensity and calculate the light energy utilization rate (LUE) and water utilization rate (WUE).

### 4.4. Plant Materials

In mid-August 2020, a 50 cm × 50 cm quadrat was randomly selected within each 2 m × 2 m quadrat for investigation, ensuring that the distance from the edge region was greater than 50 cm to avoid the influence of edge effects. During investigation of the sample plots, the aboveground biomass was clipped, and samples were screened and sorted into the functional groups of grasses and forbs. The grass functional group primarily consisted of Poaceae species including *Elymus nutans* Griseb, *Agrostis hugoniana* Rendle, and *Poa pratensis* L. All the grass plants express the Lsi1/Lsi2 transporter protein and are classified as silicon-accumulating plants. The forb functional group mainly comprised *Oxytropis ochrocephala* Bunge (Fabaceae), *Ranunculus tanguticus* (Maxim.) Ovcz. and *Anemone rivularis var. flore-minore* Maxim (Ranunculaceae), *Potentilla fragarioides* L. (Rosaceae), and *Pedicularis kansuensis* Maxim. (Scrophulariaceae)—these families lack active Si transporters, so they do not belong to silicon-accumulating plants. Samples were packed in envelopes and brought back to the laboratory. The stems and leaves were separated and dried in an oven at 70 °C to a constant weight, and the aboveground biomass was weighed and recorded. Then, the dry leaf samples were ground through a 0.150 mm sieve, packed in a self-sealing bag and labeled for determination of the contents of plant leaf nutrients and structural carbon components.

During the collection of plant samples, a soil drill with a 5 cm diameter was used to extract soil from three different locations within each 50×50 cm quadrat at a depth of 0–15 cm. The soil samples were mixed thoroughly, placed in a ziplock bag, and brought back to the laboratory. After air-drying at room temperature for 30 days, the soil was sieved through a 0.149 mm mesh. The samples were then labelled and stored in bags for subsequent analysis of soil factors.

### 4.5. Determination of Soil Characteristics

Soil pH was measured using a pH meter (PHS-5, Shanghai AiCe Electronics, Shanghai, China) with calibration against pH 4.01 and 7.01 buffers at a 1:2.5 soil-to-water ratio. Organic matter was determined by the potassium dichromate oxidation method (Walkley–Black), with titration of excess Cr_2_O_7_^2−^ using FeSO_4_. Total N was analyzed via sulfuric acid digestion (H_2_SO_4_-K_2_SO_4_-CuSO_4_-Se catalyst), followed by dilution and measurement on a SmartChem200 analyzer (WestCo Scientific Instruments, Brookfield, CT, USA). Inorganic N (NH_4_^+^-N and NO_3_^−^-N) was extracted with 2 M KCl (5:1 water-to-soil ratio) and quantified on the SmartChem200. Total P was digested with HClO_4_-H_2_SO_4_, neutralized, and measured on the SmartChem200. Available P was extracted using Bray-1 solution (0.03 M NH_4_F + 0.025 M HCl) and determined by molybdenum-blue colorimetry at 880 nm. Total silicon (Si) was analyzed via alkali fusion (NaOH at 750 °C), dissolved in 6 M HCl, and measured spectrophotometrically at 812 nm using the silicon-molybdenum blue method.

### 4.6. Leaf C, N, P, and Si Contents

Total C in plant leaves was determined by the external heating method with potassium dichromate. The content of total N and total P in leaves was determined by the HClO_4_-H_2_SO_4_ method. The Si concentration in plant tissues was determined using the method described by [64]. The Si:C, Si:N, and Si:P ratios were calculated on the basis of the contents of Si, C, N, and P.

### 4.7. Lignin, Cellulose, Hemicellulose, and Total Phenol Contents

Leaf structural carbon components were extracted according to the method of Foster et al. [65]. The lignin content was determined according to Brinkmann et al. [66]. Cellulose was extracted with trifluoroacetic acid and oscillated for 2 h at 99 °C in a 50× *g* shaker. The tubes were then centrifuged at 20 °C at 17,000× *g* for 10 min, after which the liquid phase was discarded. The cellulose content was determined using the method of Foster et al. [65]. Hemicellulose was calculated as the difference between the sum of cellulose and lignin. The total phenol was measured using folinol colorimetry. Folin–Ciocalteu reagent was added and mixed with the sample. After reaction for 20 min, the total phenol content was determined at 765 nm with a spectrophotometer (Puxin T6 New Century, Beijing Purkinje General Instrument Co., Ltd., Beijing, China).

### 4.8. Statistical Analysis

In this study, Microsoft Excel 2021 was used for data processing and analysis, and Origin 2021 software was used for mapping. The effects of the various fertilization treatments on grasses and forbs were analyzed by one-way analysis of variance (ANOVA). Two-factor analysis of variance (two-way ANOVA) was used to determine the interaction effects of P and Si on grass and forb plants. The paired sample *t* test was used to analyze the difference between the Si-added and Si-free treatments. Pearson correlation analysis was used to analyze the relationships between indices. Principal component analysis (PCA) at the different P and Si levels was performed to elucidate the interrelationships between the major dependent variables closely associated with grasses and forbs.

## 5. Conclusions

In alpine meadows, both P and Si additions increased the biomass of grasses and Forbs. Furthermore, Si addition significantly reduced structural carbon components in the leaves of both forbs and grasses, confirming the hypothesis that Si can substitute for structural carbon compounds in plants. On one hand, the lack of a clear correlation between Si and C in both grasses and forbs suggests that the C saved by Si substitution of structural compounds was reallocated to photosynthesis, increasing the Pn, thereby generating more photosynthetic products allocated to growth. On the other hand, Si addition moderately enhanced N and P content in both forbs and grasses, reflecting that increased Si levels can optimize the photosynthetic process in leaves, enabling them to sustain higher biomass. In conclusion, Si can replace structural carbohydrates and regulate the absorption and utilization of N and P to optimize the photosynthetic process of leaves, thereby achieving greater biomass.

## Figures and Tables

**Figure 1 plants-14-01779-f001:**
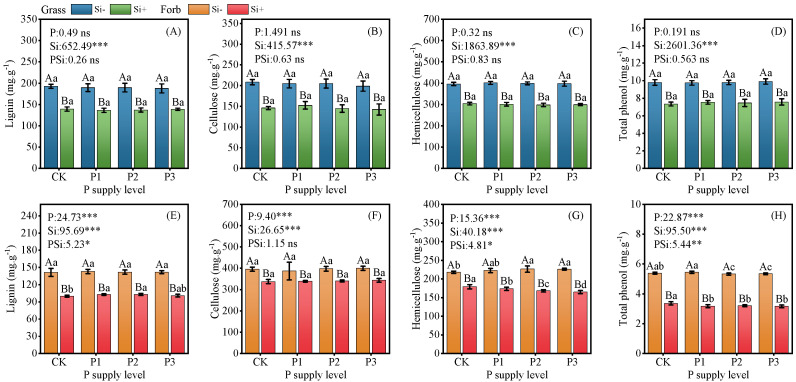
Effects of phosphorus and silicon additions on the structural carbon component in grass and forb leaves. (**A**) lignin of grass, (**B**) cellulose of grass, (**C**) hemicellulose of grass, (**D**) total phenol of grass, (**E**) lignin of forb, (**F**) cellulose of forb, (**G**) hemicellulose of forb, (**H**) total phenol of forb. Values are means of six replicates ± SD. The data were analyzed by two-way ANOVA conducted with P and Si as sources of variation. The significance of the sources of interaction (P × Si) was determined through the *p*-values: ns, not significant; * *p* < 0.05; ** *p* < 0.01; *** *p* < 0.001. Lowercase letters indicate significance between different phosphorus treatment levels (*p* < 0.05), while uppercase letters indicate significance between silicon-added and silicon-removed treatments at the same phosphorus level (*p* < 0.05).

**Figure 2 plants-14-01779-f002:**
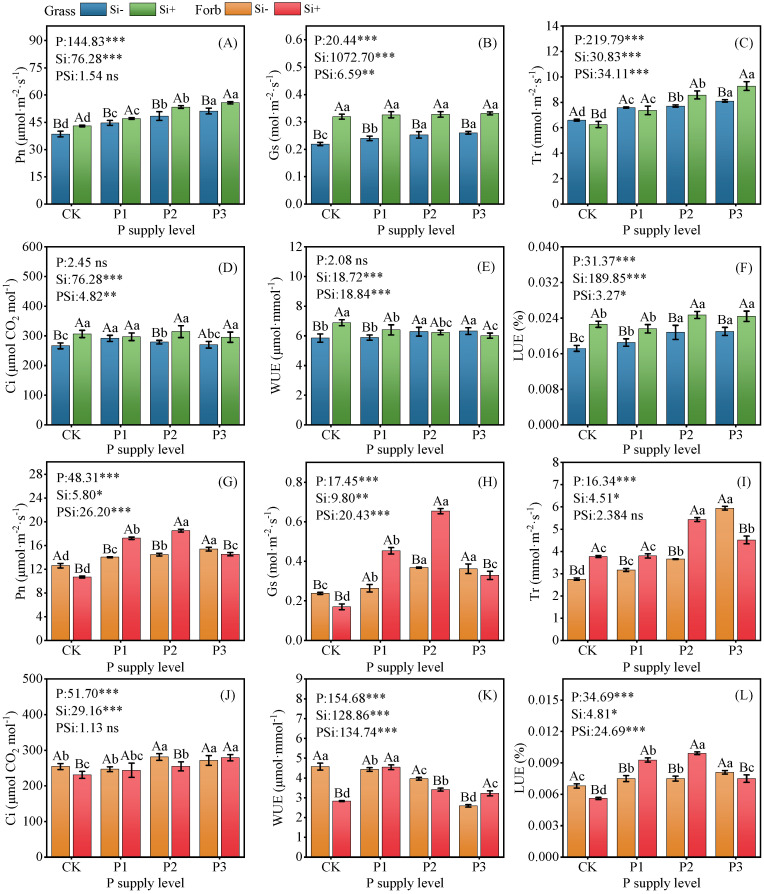
Effects of phosphorus and silicon addition on the photosynthetic performance of grass and forb leaves. (**A**) Pn of grass, (**B**) Gs of grass, (**C**) Tr of grass, (**D**) Ci of grass, (**E**) WUEof grass, (**F**) LUE of grass, (**G**) Pn of forb, (**H**) Gs of forb, (**I**) Tr of forb, (**J**) Ci of forb, (**K**) WUE of forb, (**L**) LUE of forb. Values are means of six replicates ± SD. The data were analyzed by two-way ANOVA conducted with P and Si as sources of variation. The significance of the sources of interaction (P × Si) was determined through the *p*-values: ns, not significant; * *p* < 0.05; ** *p* < 0.01; *** *p* < 0.001. Lowercase letters indicate significance between different phosphorus treatment levels (*p* < 0.05), while uppercase letters indicate significance between silicon-added and silicon-removed treatments at the same phosphorus level (*p* < 0.05).

**Figure 3 plants-14-01779-f003:**
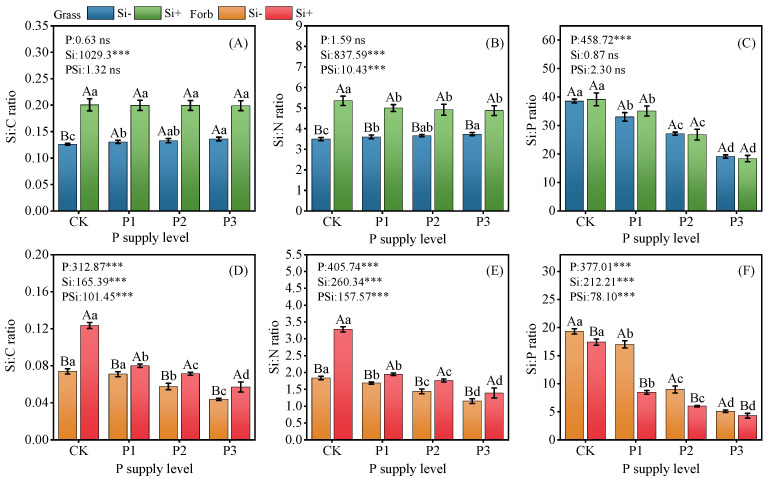
Effects of phosphorus and silicon addition on the stoichiometric ratios of C, N, and P in grass and forb leaves. Values are means of six replicates ± SD. (**A**) Si:C of grass, (**B**) Si:N of grass, (**C**) Si:P of grass, (**D**) Si:C of forb, (**E**) Si:N of forb, (**F**) Si:P of forb. The data were analyzed by two-way ANOVA conducted with P and Si as sources of variation. The significance of the sources of interaction (P × Si) was determined through the *p*-values: ns, not significant; *** *p* < 0.001. Lowercase letters indicate significance between different phosphorus treatment levels (*p* < 0.05), while uppercase letters indicate significance between silicon-added and silicon-removed treatments at the same phosphorus level (*p* < 0.05).

**Figure 4 plants-14-01779-f004:**
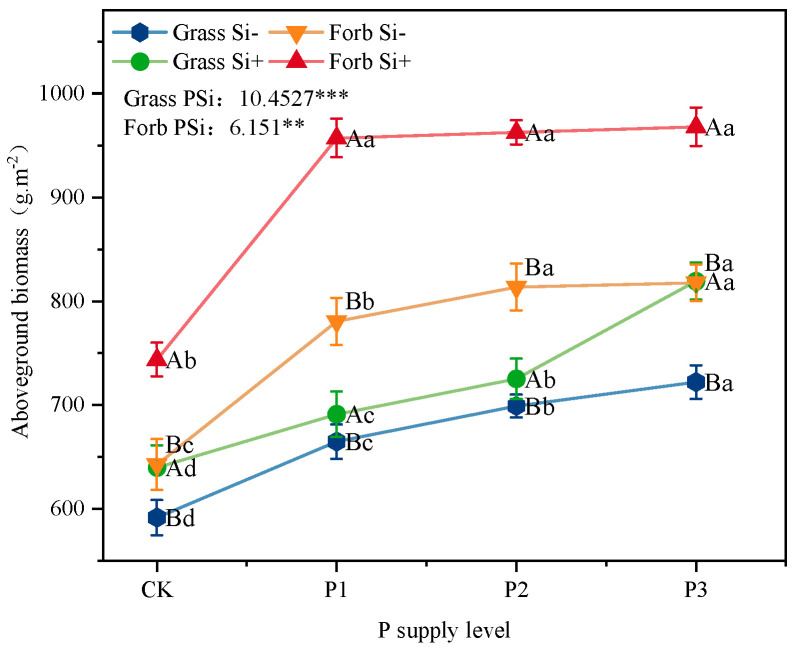
Effects of phosphorus and silicon addition on grass and forb biomass. Values are means of six replicates ± SD. The data were analyzed by two-way ANOVA conducted with P and Si as sources of variation. The significance of the sources of interaction (P × Si) was determined through the *p*-values: ns, not significant; ** *p* < 0.01; *** *p* < 0.001. Lowercase letters indicate significance between different phosphorus treatment levels (*p* < 0.05), while uppercase letters indicate significance between silicon-added and silicon-removed treatments at the same phosphorus level (*p* < 0.05).

**Figure 5 plants-14-01779-f005:**
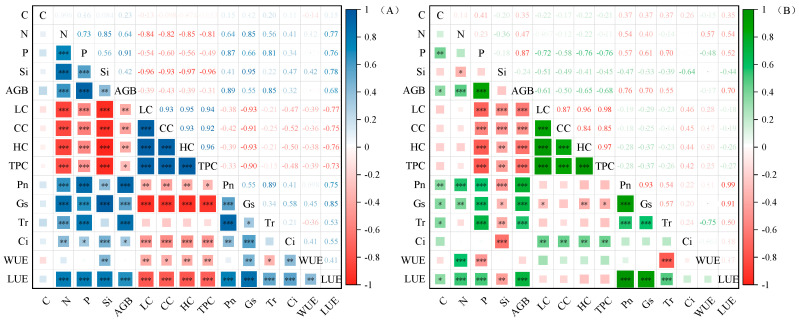
Correlation between leaf photosynthetic performance and structural carbohydrates in grasses (**A**) and forbs (**B**). Note: blue and green indicate positive correlations; red indicates negative correlations; * (*p* < 0.05) ** (*p* < 0.01) *** (*p* < 0.001). The intensity of the color indicates the importance of the variable. AGB, aboveground biomass; LC, lignin content; CC, cellulose content; HC, hemicellulose content; TPC, total phenol content; Pn, net photosynthetic rate; Gs, stomatal conductance; Tr, transpiration rate; WUE, water use efficiency; LUE, light energy utilization rate; Ci, intercellular carbon dioxide concentration.

**Figure 6 plants-14-01779-f006:**
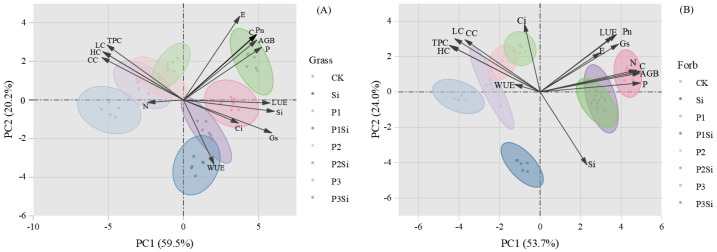
PSi treatment and PCA of grasses and forbs (**A**,**B**). Note: AGB, aboveground biomass; LC, lignin content; CC, cellulose content; HC, hemicellulose content; TPC, total phenol content; Pn, net photosynthetic rate; Gs, stomatal conductance; E, transpiration rate; WUE, water use efficiency; LUE, light energy utilization rate; Ci, intercellular carbon dioxide concentration.

**Figure 7 plants-14-01779-f007:**
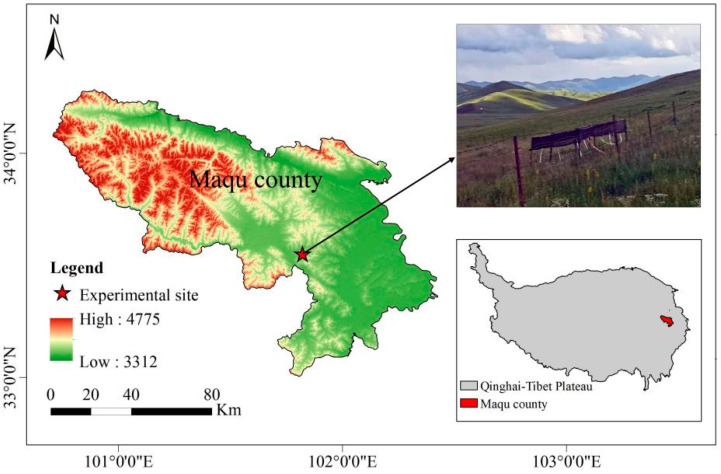
Geographical location of the research site in Maqu County, Gannan Autonomous Prefecture, Gansu Province.

## Data Availability

The data are available from the corresponding author on reasonable request.

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
