# Peer review of "Silicon Reduce Structural Carbon Components and Its Potential to Regulate the Physiological Traits of Plants"

_plants, 2025, doi:10.3390/plants14121779_

Round 1
Reviewer 1 Report
Comments and Suggestions for Authors
Dear Authors
For me, the main issue with this manuscript is that it does not specify (and probably does not even mention!) the study species, let alone differentiate them. Grasses are Si hyperaccumulators, whereas forbs are a general term that includes a wide range of species from Si hyperaccumulators to Si excluders. You cannot simply put all of them in the same bag. Perhaps some of the trends you see are not a result of P/Si treatments on discrete species, but of changes in community composition and interspecific interactions (e.g., grasses overshadowing other species).
In the Results, make sure it is always clear what are the treatments you refer to and what are the directions of trends (i.e., "P application increased LUE", not "P had an effect on LUE").
The level of writing (structure, not vocabulary or grammar) is overall fair, but often does not meet common standards. For example, in the Introduction there are sharp breaks between the first two and last two paragraphs, where it seems as if you jump from one topic to another.
L 17: You do not define PSi.
L 20-22: You should elaborate more on the underlying mechanism/explanation.
L 23-25: Above, you never mention any patterns (results) relating to P deficiency or excess.
L 35-37: What about P excess?
L 39-40: There are three types of plants in this respect: active uptake, passive uptake and exclusion. There are many important differences between active and passive uptake, and you need to mention this here. This is also a good place to note that grasses are typical Si hyperaccumulators through active uptake.
L 45-50: You should mention that Si effects on P absorption include both soil processes and in planta processes, and give 1-2 examples of each.
L 51-52: I suggest citing papers such as Hodson & Guppy, 2022, Plant and Soil and Katz et al., 2024, Physiologia Plantarum. You should also explain why such a trade-off exists or is postulated. As a more general comment, I suggest avoiding writing "to the best of our knowledge", since it implies you may not have reviewed the literature thoroughly enough.
L 54-58: Note also a more thorough and critical discussion of the matter in de Tombeur et al., 2023, Trends in Ecology and Evolution.
L 67-69: Joerg Schaller did a lot of work on the topic in other ecosystems.
L 73-74: If you don't measure energy consumption, don't hypothesise about it.
L 72-76: Hypotheses should be stated as truths, without "may". The idea is to test them, and then use "may" in the conclusions if appropriate.
L 79-85, 99-101, 116-120: These are not results.
L 86-87: Under what conditions?
L 87: Friendly advice: always write exact P values. P < 0.05 can be 0.001 or 0.049, and there is great difference between the two. Likewise, P > 0.05 can be 0.051 or 0.90. Finally, 0.049 is significant, whereas 0.051 is insignificant, although the difference between them may be due to how you round numbers when doing your measurements.
Figure 1: Clarify what different letters mean, especially since you have two sets (uppercase and lowercase), what levels of significance each number of asterisks represents, and if error bars are standard deviation or standard error.
L 101-103: Where can I see this?
Figure 3: You should also show the concentrations of each element separately.
Figure 5: Using blue and green or red and orange for contrasting trends is confusing. Use contrasting colours.
Figure 6a, in my opinion, says nothing interesting. Of course grasses and forbs are not the same thing.
L 178-184, 202-205: This belongs in the Introduction.
L 190: Does this reduce costs of biosynthesis, or simply divert costs from biosynthesis to Si uptake?
L 192: Why "similar"? Do the two operate the same way?
L 210: "silica-cuticle double layer" – there is not such material as "cuticle silica".
L 287-288: What about soil P content, pH, etc.?
L 357-359: You don't really show this.
Author Response
请参阅附件

Reviewer 2 Report
Comments and Suggestions for Authors
Dear Authors,
Your manuscript is very interesting and partially fills the gap of missing information about silicon importance in plants (mostly grasses) nutrition. I have some remarks and comments, which can lead to the improvement of your manuscript quality.
Lines 31, 39, and 246: missing spaces in front of references
Lines 58-60: I do not understand this sentence as well as the reference placement in the middle of this sentence.
lines 86 and 87: Here is the statement that "The contents of lignin, cellulose, hemicellulose and total phenol in grasses decreased by 28%, 30%, 23% and 25%, respectively." - I do not understand the context of this sentence. The decrease should be related to something. The same comment I have for the sentence for forbs on the lines 89-90.
Lines 107: replace ",and these..." with ",where..."
Line 139: round the numbers 819.28 and 967.97 on whole numbers (819 and 968)
Line 144: change the order "index correlation" on "correlation index"
Lines 189 -190: You confirmed the hypothesis that Si can replace structural carbon compounds. However, how did you determine that this process reduces the costs associated with biosynthesis? Can you please provide some quantification of energy costs saving or improve the discussion of this topic? How much energy costs the plant to incorporate Si in structural carbon compounds?
Line 196: Missing number of reference Tamai and Ma
Discussion: Can you please discuss the impact of Si fertilizing on potential nutritional value of forage crops. You are mentioning the possible negative effect on lines 269-270, but I think that this part of discussion should be more specific. You can describe, which specific nutrients can be affected by Si fertilizing as well as the impact of structural compounds changes on forage quality (fiber content etc.).
Line 322: What is meaning the "100 mesh"? You should add the size unit (mm, μm or other)
Methodology:
In the case of your manuscript I recommend to shift the methodology in front of results (if the editors will agree). I know that journal is requesting the order: Introduction, results, ....., however the results are following the methodology and in the methodology section are described all the measured parameters and statistical evaluation.
Line 354: you can delete "in plant"
Yours sincerely
Reviewer
Reviewer 3 Report
Comments and Suggestions for Authors
Dear author
The article presents interesting results, of interest to the scientific community, but lacks a lot of information in the methodology so that a more detailed analysis of the results can be made.
- Lacks a history of the area
- A rainfall graph
- A detailed chemical and physical analysis of the soil
- What planting and top dressing was carried out in the area?
- Has any soil acidity been corrected?
Without this information and others included in the text, it is difficult or even impossible to analyze the results.

Reviewer 4 Report
Comments and Suggestions for Authors
The manuscript "Silicon regulation structural carbon components and its potential to mitigate deficiency or excess phosphorus" present a research on P and Si addition in grassland biomes, with a gradient of fertilizer that target the identification of most suitable solutions for maintaining the stability of these ecosystems.
The aim and the hypotheses are clear and are designed to answer multiple research questions related to both plant structural components and the potential biomass from treated grasslands.
Results section - the authors present multiple complex data and significant differences between treatments. The text related to figures need to be expanded to explore completely the differences between treatments.
The Discussion section links well the findings of this study with the international literature in the field. This section will benefit from the improvements in the Results section.
Materials and Methods - this section describes the filed area, the experimental design and the methods used for data analysis.
Round 2
Reviewer 1 Report
Comments and Suggestions for Authors
The authors has responded to all my comments in a satisfactory manner.
Reviewer 3 Report
Comments and Suggestions for Authors
Dear Author ,
All of my questions and suggestions for improvements to the text were answered or accepted by the author, so I am satisfied with the corrections made by the authors, which have significantly improved the quality of the text.
The reviewer